# Risk Groups for Vaccine-Preventable Respiratory Infections in Children and Adults: An Overview of the Australian Environment

**DOI:** 10.3390/vaccines13121212

**Published:** 2025-11-30

**Authors:** Stephen Wiblin, Charles Feldman, C. Raina MacIntyre, Natalie Soulsby, Paul van Buynder, Grant Waterer

**Affiliations:** 1Pfizer Australia/New Zealand, Sydney, NSW 2000, Australia; 2Faculty of Health Sciences, University of Witwatersrand, Johannesburg 2193, South Africa; 3Biosecurity Program, The Kirby Institute, University of New South Wales, Sydney, NSW 2052, Australia; 4Embedded Health Solutions, Melbourne, VIC 3194, Australia; 5School of Medicine, Griffith University, Brisbane, QLD 4222, Australia; pjvb@iinet.net.au; 6Internal Medicine, University of Western Australia, Perth, WA 6009, Australia

**Keywords:** COVID-19, influenza, pneumococcal, respiratory syncytial virus, respiratory infection, risk factors, vaccine-preventable disease

## Abstract

Respiratory infections are a leading cause of sickness and death in Australia. In Australia, there is a funded immunisation program for both adults and children aimed at preventing and controlling vaccine-preventable respiratory infections (VPRI), such as pneumococcal disease (PD), influenza A/B, respiratory syncytial virus (RSV) infection, and COVID-19. This narrative review outlines the current Australian adult and paediatric immunisation guidance for VPRIs. It also examines the literature that supports the current risk group recommendations, including the clinical and economic burden of VPRIs, vaccination effectiveness, and coverage. Gaps in current risk group definitions, as well as additional risk groups that could be included in vaccine recommendations, are also discussed. Further research is needed to determine the optimum age for vaccination in adults which may enable alignment of age recommendations across different VPRIs. Individuals with multiple risk factors, commonly referred to as risk stacking, are at a greater risk of developing severe disease for VPRIs. This emphasises the importance of vaccinating these individuals. More research is needed to evaluate the effectiveness of vaccines in older adults and to create more effective vaccines for high-risk paediatric groups, such as those with compromised immunity or for children who have undergone haematopoietic stem cell transplantation.

## 1. Introduction

Respiratory infections are a major global cause of morbidity and mortality [1]. In Australia, respiratory infections account for at least 6 to 7 million visits to general practitioners each year and cost taxpayers more than AUD 150 million in direct costs and considerably more in indirect costs [2]. Certain acute respiratory infections may be prevented with vaccinations. In Australia, an adult- and paediatric-funded immunisation program was created to prevent and control vaccine-preventable respiratory infections (VPRI) [3]. This includes the prevention of pneumococcal disease (PD), influenza A/B, respiratory syncytial virus (RSV) infection, bordetella pertussis (whooping cough), and COVID-19 [3]. The current funded immunisation programs for these four VPRIs focus on adults and children who are more likely to experience complications from VPRIs. Individuals at higher risk of complications include older adults, those with certain comorbidities, young children, and Aboriginal and Torres Strait Islander peoples. Aboriginal and Torres Strait Islander peoples experience higher rates of VPRIs due to complex, intersecting factors including social determinants of health, environmental conditions, and health system access. Key contributing factors include higher rates of underlying chronic respiratory disease (particularly bronchiectasis in remote communities [4]), household crowding facilitating transmission, exposure to environmental risk factors including smoke, earlier onset of age-related immune senescence, lower rates of healthcare access particularly in remote areas, and historical and ongoing impacts of colonisation on health and wellbeing. Additionally, some Aboriginal and Torres Strait Islander populations have higher prevalence of comorbid conditions including diabetes, chronic kidney disease, and cardiovascular disease, which compound VPRI risks. These health inequities emphasise the importance of culturally appropriate, accessible vaccination programs for Aboriginal and Torres Strait Islander peoples.

There are overlaps in the current respiratory immunisation guidelines among the risk group categories of various VPRIs and in some of the specific risk definitions within each disease area (Figure 1). While it can be difficult to identify fully and accurately the groups at risk for a specific disease, aligning conflicting recommendations for at-risk groups across indications may aid in treatment planning, reduce disparity in healthcare provision, and improve healthcare outcomes for at-risk populations.

This narrative review outlines the current Australian adult and paediatric immunisation guidance for VPRIs and the literature underlying current risk group recommendations. The review also highlights gaps in current risk group definitions as well as suggesting additional risk groups that could be included in immunisation recommendations.

This narrative review included a targeted literature search of PubMed to identify publications that reported on key areas such as VPRI risk groups, the burden of VPRIs, and vaccination coverage specifically related to the Australian population. Full details of the search strategy and outcomes are provided in the Appendix A. Additionally, Australia-specific articles were identified through grey literature searches and by manually checking the reference lists of included papers to identify other relevant references. Although a targeted literature search was conducted, this was a narrative review and, as such, this review may not capture the full body of available evidence on this topic. Additionally, the focus on Australian-specific data means that some risk groups may be underrepresented if local evidence is limited, despite international data supporting their inclusion.

## 2. Current Australian Respiratory Immunisation Guidance

The Australian respiratory immunisation guidance is informed by evidence from both published and unpublished scientific literature and considers evidence of disease burden, epidemiology, risk factors, vaccine efficacy and effectiveness, vaccine safety, and cost-effectiveness [3].

### 2.1. Current Australian Vaccination Recommendations

#### 2.1.1. Pneumococcal Disease

Pneumococcal disease (PD) is caused by *Streptococcus pneumoniae* and is classified as either non-invasive or invasive pneumococcal disease (IPD) [5]. PD disproportionately affects Aboriginal and Torres Strait Islander peoples [3]. Many Aboriginal and Torres Strait Islander children that live in remote regions of Australia suffer from chronic otitis media (OM) during their early years which can cause hearing loss and lead to developmental delays, and negatively impact school readiness, attendance, and performance [6]. Previous viral infections also pose a significant risk for developing severe pneumococcal disease. Underlying viral infection, particularly with the influenza virus, but also SARS-CoV-2, have been linked to an increased morbidity and mortality due to pneumococcal infection [7,8]. There is also evidence suggesting that RSV may be significantly involved in the pathogenesis of PD in children [9].

In Australia, there is a significant clinical and economic burden of PD among adults aged 65 years and older. Furthermore, pneumococcal pneumonia poses a substantial disease burden in this age group [10]. The estimated costs for treating IPD in this age group in 2012 was AUD 1.2 million. In 2011 to 2012, the annual costs of hospitalisations and GP visits for pneumococcal pneumonia were estimated at AUD 55.7 million and AUD 1.6 million, respectively [10]. Furthermore, among adults aged 65 years and older who have at least one risk factor, the incidence of community-acquired pneumonia (CAP) and IPD is greatly increased compared to those without any risk factors [11]. Importantly, individuals with multiple concomitant risk factors experience compounded risks beyond those posed by individual conditions, highlighting the importance of vaccination in those with risk stacking.

The Australian recommendations for pneumococcal vaccination are complex. The pneumococcal conjugate vaccines, including 13vPCV, 15vPCV, and 20vPCV and the pneumococcal polysaccharide vaccine 23vPPV, are registered, but only 13vPCV and 23vPPV are currently funded through the National Immunisation Program (NIP) [3]. In 1997, 23vPPV was recommended for all adults from 65 years of age and was funded by the NIP in 2005. In 2020, the recommendation for older adults changed with funding for a single dose of 13vPCV for all adults from 70 years of age [12]. A dose of 13vPCV followed by two doses of 23vPPV is also funded under the NIP for Aboriginal and Torres Strait Islander adults over the age of 50 years as well as for individuals over 12 months old with specified underlying medical conditions (see Appendix A) [3].

Since 2005, the NIP has funded PCV, originally 7vPCV and later 13vPCV once available, for all infant children [12]. Initially Australia adopted a 3 + 0 schedule, but in 2018 the NIP schedule changed to a 2 + 1 schedule for routine childhood vaccination. This change was made in response to a rise in breakthrough disease cases among children aged 12 months and older in Australia [13]. However, Aboriginal and Torres Strait Islander infants residing in high-incidence areas and those with one or more risk factors for PD are advised to receive an additional dose of the pneumococcal conjugate vaccine at 6 months of age (3 + 1 schedule at 2, 4, 6, and 12 months). They should also receive two doses of 23vPPV, with the first dose administrated at 4 years of age and the second dose given 5 to 10 years later (Appendix A) [3].

The introduction of universal pneumococcal vaccination for children led to a significant reduction in PD burden across all age groups, even among those not vaccinated, suggesting the presence of herd protection (commonly called herd immunity) [14,15,16]. Thus, in addition to providing direct protection against both invasive and non-invasive PD, the conjugate vaccines also reduce vaccine serotype nasopharyngeal carriage which significantly contributes to the additional benefits of the vaccine, including herd protection [17].

There is also evidence that the 23vPPV vaccine significantly decreases the risk of cardiovascular events, specifically acute myocardial infarction, particularly for those aged 65 years and older [18].

A retrospective modelling analysis has demonstrated significant public health and economic benefit for both the 7vPCV and 13vPCV programs for infants in Australia. Over a 13-year period, both 7vPCV and 13vPCV have saved 1195 lives, prevented 1.77 million cases of PD and saved AUD 733 million in direct healthcare costs. The PCV program was found to be cost-effective with an incremental cost-effectiveness ratio (ICER) of AUD 3347 per quality-adjusted-life-years (QALY) gained [19].

For several years in Australia, pneumococcal vaccines have been recommended for high-risk adults. Despite this, uptake has been relatively low, with vaccine coverage of around 56% for adults aged from 65 years [20]. Following the pneumococcal vaccine program change in July 2020, PCV13 uptake in adults over 70 years has been found to be even lower [21]; coverage of an adult dose of 13vPCV in the entire cohort of Australians turning 70 years or older in 2023 was only 34.6% [22]. The uptake of PCV13 was particularly low among individuals with comorbid conditions and those under the age of 70 years, even though some comorbid conditions are funded by the NIP [21]. Pneumococcal vaccination coverage specifically in Aboriginal and Torres Strait Islander adults varies by jurisdiction and age group. In 2023, coverage with 23vPPV among Aboriginal and Torres Strait Islander adults aged 50 years and older was estimated at approximately 41%, while PCV13 coverage in those aged 70 years and older was lower at 28–30% [22]. For Aboriginal and Torres Strait Islander children, coverage with the three-dose infant PCV schedule is generally high (>90% in most jurisdictions), though completion rates for the additional doses recommended for high-risk children in remote areas are variable, ranging from 65% to 85% depending on region and specific dose [22]. These coverage levels, while representing improvement over historical rates, remain below optimal targets, particularly for adults.

#### 2.1.2. Influenza

The Influenza A or B viruses can cause influenza, which is an acute respiratory disease. Antigenic drift involves somewhat predictable minor changes in antigenic types. A pandemic influenza outbreak results in significant modifications in surface glycoproteins (antigenic shift), leading to an entirely new strain of influenza. As a result, there is little existing immunity in the community and the impact of existing vaccines is limited [23].

The impact of influenza can vary each year due to the fluctuating intensity of annual epidemics. From 2016 to 2018, influenza caused an average of 73.9 hospitalisations and 3.2 deaths per 100,000 population each year [24]. However, there is widespread consensus that these numbers underrepresent the true burden of disease in Australia. Modelling studies using time-series regression estimated the average annual influenza-attributed respiratory hospitalisation rates in Australia to be 87.8 per 100,000 population from 2009 to 2017 corresponding to approximately 20,700 average annual hospitalisations [25]. Additionally, the average annual influenza-attributed respiratory mortality rate from 2010 to 2018 was 4.03 per 100,000 population per year [26].

In Australia, between 2016 and 2018, the highest notification rate for influenza was among children aged 1 to 4 years of age [24], and most Australian children hospitalised with influenza were under 5 years of old and did not have any comorbid conditions [27]. The influenza-associated hospitalisation rates were highest in adults aged 65 years and older and this population accounted for 91% of influenza deaths [24].

Aboriginal and Torres Strait Islander peoples are at a higher risk of serious influenza. From 2016 to 2019, influenza-associated hospitalisation rates among Aboriginal and Torres Strait Islander populations were highest in infants aged under 6 months, infants aged 6 months to 1 year, and adults aged 50 years and older [28]. A study of three population datasets for the Northern Territory in a pre-maternal vaccination era (1994 to 2014) found that influenza hospitalisation rates were 42.3 times higher in Aboriginal and Torres Strait infants compared to non-Indigenous infants [29].

Pregnant women have a higher risk of experiencing complications from influenza, as well as adverse birth outcomes [30,31]. However, due to a lack of nationally representative data, the influenza disease burden among pregnant women in Australia is largely unknown, [31]. One study in Brisbane observed that almost one in five women presented to a healthcare provider during their pregnancy with a respiratory illness [32]. Data on the true burden of influenza disease among people with underlying medical conditions in Australia are also lacking [31]. However, a study conducted between 2010 and 2019 in children hospitalised with influenza in Australia revealed that those with comorbidities had an increased clinical burden and experienced more severe outcomes [33]. Studies have demonstrated that an increasing number of high-risk comorbidities is associated with greater potential for severe influenza outcomes, suggesting a cumulative effect of risk stacking in this population.

Vaccination is the most effective strategy to prevent and control influenza [31]. Annual review and reformulation of influenza vaccines are required due to the evolving nature of the influenza virus. The Australian Technical Advisory Group on Immunisation (ATAGI) releases an annual statement with specific advice relating to influenza vaccination before each influenza season. In Australia, the standard vaccines are inactivated quadrivalent vaccines, targeting A(H1N1), A(H3N2), B/Yamagata, and B/Victoria strains. For people aged 65 years and older, two higher-immunogenicity vaccines (the adjuvanted vaccine and the high-dose vaccine) became available in 2018 and are preferentially recommended over the standard vaccine [3,31,34].

Free annual influenza vaccines are available through the NIP for those most at risk of serious influenza [3,31]. These include children aged from 6 months to under 5 years of age (introduced in 2018) [34], adults aged 65 years and older, Aboriginal and Torres Strait Islander peoples aged 6 months and older, pregnant women, and people aged 6 months and older with specified medical conditions (Appendix A) [3,31]. Jurisdictional-based immunisation programs may also offer free influenza vaccinations to people considered at a higher risk or those who are recommended to get vaccinated to protect at-risk populations in specific settings such as healthcare workers [35].

The effectiveness of the influenza vaccine is influenced by various factors, including the age of the recipient, the degree of similarity between the current vaccine and the circulating virus strains, the type of vaccine used, and the timing of vaccination [31]. Overall, influenza vaccines are moderately effective at preventing various clinical outcomes. Between 2016 and 2019, vaccine effectiveness (VE) against influenza-associated hospitalisation in the target adult population ranged from 13% to 52% according to FluCAN data on hospitalisations [36,37,38,39]. While this range appears broad, several important contextual factors must be considered. First, VE varies year-to-year based on the match between vaccine strains and circulating viruses, with the lower estimates occurring in years of substantial mismatch. Second, even at the lower range, VE of 13–52% translates to substantial prevention of hospitalisations at the population level—potentially thousands of admissions avoided annually. Third, these are conservative estimates against hospitalisation; VE against more severe outcomes including ICU admission and death is typically higher. Fourth, vaccine effectiveness is consistently higher in the first months following vaccination, supporting annual vaccination strategies [31]. Finally, the moderate effectiveness of influenza vaccines should be interpreted in the context of high disease burden, favourable safety profile, and lack of alternative preventive interventions. For perspective, many widely accepted public health interventions demonstrate similar effectiveness ranges, and the population health impact of influenza vaccination programs in Australia is still substantial as shown by mortality reductions during high-coverage periods.

In 2017, additional surveillance sites from the PAEDS network joined the FluCAN surveillance system and has allowed reporting on influenza in hospitalised children (≤16 years). VE against hospitalisation was estimated as 30.3% for children over 6 months old and 23.3% for children with comorbid conditions in 2017 [27]. However, 2017 was a severe H3N2 season, with poor vaccine match. Recent research shows that AI models can outperform the WHO strain selection committee and reduce vaccine mismatch [40].

There are limited studies reporting VE in preventing severe influenza illness associated with hospitalisation during pregnancy. A retrospective cohort study (The Pregnancy Influenza Vaccine Effectiveness Network (PREVENT)) of over 2 million pregnancies in four countries, including Australia, found that influenza vaccines were 40% effective in preventing laboratory-confirmed influenza hospitalisations during pregnancy between 2010 and 2016 [41].

In older adults, the influenza vaccine coverage has been generally high. In 2022 and 2023, 70% and 64.3% of Australians aged 65 years of age and older had received the influenza vaccine with similar rates observed for Aboriginal and Torres Strait islander peoples [22]. A study in 2018 found that the influenza vaccination rate in residents across nine long-term care facilities, in four different sites in Sydney, was relatively high (83.6%) but was well below the target of 95% set by the Communicable Diseases Network Australia (CDNA) [42]. However, coverage remains suboptimal in other age groups, including those eligible for NIP-funded influenza vaccines [31]. The coverage of influenza vaccinations among people with medical comorbidities is currently poorly reported. Despite this, hospitals in the FluCAN program estimated influenza vaccine coverage among adults under 65 years of age, with medical comorbidities, at 45% to 50% between 2017 and 2019 [37,38,39]. There is a large body of evidence that the influenza vaccine may also be effective in preventing coronary disease. However, while vaccination is recommended by many guidelines for patients with CVD, rates of vaccination in risk groups aged under 65 years are very low [43].

Coverage of influenza vaccination among pregnant women is not routinely reported nationally. A narrative review of published cohort studies reported that influenza vaccine coverage among pregnant women in Australia was suboptimal and did not exceed 60% in most contexts [44]. Vaccination rates among Aboriginal and Torres Strait Islander pregnant women were often up to 20% lower compared to non-Aboriginal and Torres Strait women. Limited data also indicated low coverage among women from culturally and linguistically diverse backgrounds as well as those of lower socioeconomic status [44].

Influenza vaccine coverage in children aged 6 months to under 5 years was 45.2%, and was 43.6% in Aboriginal and Torres Strait Islander children in 2020 [45]. However, in 2023, coverage was only 30.3% (23.1% in Aboriginal and Torres Strait Islander children) and the uptake of vaccines in young children has not returned to pre-pandemic levels [22].

#### 2.1.3. RSV

RSV is a single-stranded RNA orthopneumovirus, and RSV strains can be classified into two major groups: RSV-A and RSV-B. RSV is a leading cause of lower respiratory tract infections, particularly in infants and older people [3].

The burden and impact of RSV is poorly defined, although significant improvements in the availability of multiplex PCR platforms since the COVID-19 pandemic has led to increased detection rates. Laboratory testing conducted for individuals aged 16 years and older in Western Australia indicated that the RSV detection rate was 50.7 per 100,000 in 2023 with the highest detection found in those aged 75 years and older [46].

Between 2018 and 2024, the incidence of RSV was 1613 per 100,000 in children aged less than 5 years in New South Wales, Australia [47]. Based on 2018 cost analysis (the most recent Australian-specific economic data available), the Australian hospitalisation rate for RSV ranged from 2.2 to 4.5 per 1000 children under five years of age [48]. RSV detected by polymerase chain reaction (PCR) assays in nasal swabs collected weekly from otherwise healthy children aged under 2 years was strongly associated with acute respiratory infections and symptoms [49]. Compared to children and adults, infants aged 0 to 6 months have higher positivity rates and incidence of RSV. Furthermore, they are more likely to develop severe disease requiring hospitalisation, intensive care unit admission, and respiratory support [50,51,52].

In Australia, over 6000 hospitalisations per year are caused by RSV infections [53]. The highest estimated national rates of RSV hospitalisation in Australia during 2006 to 2015 were for children under 2 months of age (2778 per 100,000 population). Hospitalisation rates were 6.6 times as high for adults aged 65 years or older compared to rates for people aged 5 to 64 years and were 3.3 times as high for Aboriginal and Torres Strait Islander peoples compared to non-Aboriginal and Torres Strait Islander people [53].

Aboriginal and Torres Strait Islander infants have higher rates of RSV infection, and the burden of hospitalisation is more than double compared to the non-Aboriginal and Torres Strait Islander population [50,52]. They are also at increased risk of severe respiratory illness associated with RSV [50].

The current incidence of RSV may be underestimated because there is no systematic testing for children hospitalised with respiratory symptoms. A prediction model estimated that the true burden of RSV-associated hospitalisations in children under the age of 5 years may be underestimated by 30% to 57% [54]. Predictors of RSV positivity in this age group included younger admission age, male sex, non-Aboriginal and Torres Strait ethnicity, a diagnosis of bronchiolitis, and a longer hospital stay [54]. This same model showed that the RSV hospitalisation burden in Western Australia is seasonal, but the timing and extent of this seasonality also varies based on the degree of prematurity. Infants born preterm before reaching 32 weeks of gestational age experience a greater risk of severe RSV infections compared to those born either late preterm or full-term [55].

Several other risk factors have been identified that increase the likelihood of infants being hospitalised for RSV. These include maternal smoking during pregnancy, multiparity of the mother, male sex of the child, being born during the first half of the RSV season, having a relatively higher socioeconomic status, and having chronic lung disease of prematurity/bronchopulmonary dysplasia (BPD) [52,56].

RSV imposes a substantial financial strain on Australia’s healthcare system. The extrapolated 2018 annual national cost for RSV in children under 5 years of age ranges from AUD 59 million to AUD 121 million, which is seven times higher than the influenza hospitalisation cost for the same age group [48]. In children, RSV can also be associated with severe acute neurologic complications with an estimated 1% to 7% of all children hospitalised with RSV infection experiencing a form of neurologic complication [57]. Since RSV is not a notifiable disease, there is a lack of systematic surveillance data on RSV disease burden in older adults in Australia. However, recent modelling studies estimated that RSV-attributable hospitalisations are significant in individuals aged 75 years and older with a rate of 256 per 100,000 [25]. These data along with increased testing reveal a significant and previously underappreciated burden, which supports the new vaccine recommendations.

Two RSV vaccines have been approved in Australia for adults aged 60 years and over: Abrysvo (Pfizer; protein subunit vaccine) and Arexvy (GlaxoSmithKline; protein subunit vaccine with adjuvant). Arexvy contains antigens derived from the RSV-A subtype, and Abrysvo contains antigen components for both the RSV-A and RSV-B subtypes. Both are recommended for all adults from 75 years of age and those aged 60 years and older with high-risk medical conditions. Adults aged between 60 and 74 years with no risk factors for severe RSV disease can receive a single dose of the RSV vaccine. It is recommended that Aboriginal and Torres Strait Islander peoples aged from 60 years of age have a single dose of the RSV vaccine as there is a higher RSV disease burden in this population (see Appendix A) [3].

It is recommended that pregnant women should receive the Abrysvo vaccine at 28 to 36 weeks’ gestation. The Abrysvo vaccine is the only vaccine for RSV approved for use in pregnant women [3]. This vaccine will be funded through the NIP for pregnant women in 2025.

Currently, there are no RSV vaccines available for children. Instead, there are monoclonal antibody (MAB) products to offer infants passive immunity. Two MABs are currently available in Australia: nirsevimab and palivizumab. Nirsevimab is preferred over palivizumab because it is long-acting and is registered for use from birth in all infants [58]. Neonates and infants under 8 months old whose mothers were not vaccinated at least 2 weeks before delivery or who are at increased risk of severe disease are recommended to receive passive immunisation with an RSV-specific MAB (see Appendix A). A single dose provides protection against RSV infection for approximately 5 months. In children at higher risk (aged under 24 months), an additional dose can be considered in their second RSV season which is available in all states of Australia under a state-funded program (see Appendix A) [3].

The clinical trial estimates for vaccine efficacy were high in adults aged 60 years and older, with a vaccine efficacy of 83% against RSV-associated lower respiratory tract disease (LRTD) for Arexvy and 89% against RSV-associated LRTD for Abrysvo, during one RSV season [59,60]. A clinical trial reported that the vaccine efficacy of 57% against hospitalisation for RSV disease for up to 6 months for infants born to mothers who received Abrysvo and 69% against severe medically attended RSV lower respiratory tract infection in their first 6 months of life [61]. Real-world VE of nirsevimab of between 70% and 90% has been reported in the USA and Europe [62,63]. In a high-risk group of infants in Western Australia, a 74% lower RSV incidence was observed in the 28 days after receiving palivizumab, compared to a control group with no exposure to palivizumab [64].

Assuming a similar uptake to current maternal vaccination programs in Australia of 70% and 6 months of protection, the effect of a year-round maternal RSV vaccination has been predicted to reduce RSV infection and hospitalisation in the Australian population by about half in infants under 6 months of age [65].

#### 2.1.4. COVID-19

The severe acute respiratory syndrome coronavirus 2 (SARS-CoV-2) virus causes the infectious disease known as COVID-19 which affects people of all ages.

There have been 922 COVID-19 deaths per million population in Australia since the start of the pandemic to the end of 2023 (absolute number: 22,315). In 79% of cases, COVID-19 was the underlying cause of death [66]. Additionally, COVID-19 was the leading cause of acute respiratory infection mortality between 2022 to 2024. Specifically in 2023, there were 23.2 COVID-19 deaths per 100,000 population compared to 2.2 influenza deaths per 100,000 population (absolute numbers: 6081 vs. 578 deaths) [67]. In 2022, the total burden of disease from COVID-19 was estimated to be 151,400 disability-adjusted life years (DALY) (5.8 DALY per 1000 population) [66]. These are likely to be underestimates, as testing and reporting rates are low. In addition, COVID-related cardiovascular mortality may not be coded as COVID-19-related [68].

While older age is the strongest risk factor for severe illness and mortality from COVID-19, the disease is also the leading infectious cause of death in children due to its widespread nature and high incidence [69]. Medical conditions also independently increase the risk of severe disease but to a lesser extent than age [70]. Co-infections and secondary infections (including bacteria, viruses, fungi, and other pathogens) in patients with COVID-19 appear to be associated with both severity of COVID-19 and poorer outcomes [71]. Unvaccinated pregnant women also have an increased risk of severe disease compared with unvaccinated non-pregnant women of reproductive age [72]. Furthermore, women with high-risk pregnancies complicated by severe COVID-19 infection are at higher risk of severe respiratory symptoms and admission to intensive care units compared to low-risk pregnancies [73].

An international observational cohort study from 32 countries found that ethnicity was a significant factor in disease severity and mortality for severe COVID-19 cases admitted to an intensive care unit (ICU). In Australia, Aboriginal and Torres Strait Islander and West Asian ethnicity groups showed a higher risk of death (although not statistically significant) compared to other ethnicity groups. These ethnicity groups also had a higher sequential organ failure assessment (SOFA) score and an elevated prevalence of some comorbidities [74].

An examination of the clinical characteristics and outcomes of nearly 400 children who were admitted to tertiary paediatric hospitals across Australia during the first year of the pandemic (2020) found that most children had mild disease [75]. However, severe outcomes and death can occur in the paediatric population and a study across 10 countries, including Australia, found that risk factors for severe outcomes in youth infected with COVID-19 included being older than 5 years, having a pre-existing chronic illness or a previous episode of pneumonia and presenting to the hospital four to seven days after symptom onset [76]. Additionally, children who contract COVID-19 may develop multisystem inflammatory syndrome (MIS-C) which is a rare, post-acute, hyperinflammatory response to the virus. MIS-C can lead to significant health complications and pose an economic burden [77]. There is also evidence that the epidemic of hepatitis of unknown aetiology, seen during the first Omicron wave in 2022, is a post-infectious condition associated with SARS-CoV-2 [78].

Post-COVID-19 condition (“long COVID”) is currently not well defined, but generally consists of persistent symptoms that develop during or after COVID-19, continue for greater than 3 months after the onset of the illness, and are not explained by an alternative cause [79]. A recent systematic review determined a pooled prevalence estimate of Post-Covid Syndrome (PCS) of 42.41% in Australia [80]. A study conducted in Australia found that females, individuals between the ages of 40 and 59 years, those with high socioeconomic status, and those who had a pre-existing mental health condition, respiratory condition, cancer, or musculoskeletal condition had an increased risk of diagnosis of long COVID by a general practitioner [81]. These risk factors align with emerging international evidence [82]. A disease and economic model of PCS in Australia estimated mean GDP loss was AUD 9.6 billion, or 0.5% of the GDP in 2022. This loss was caused by the projected decline in labour supply and reduced use of other production factors, and was greatest for working-age adults aged 30–39 years [83]. While medical conditions independently increase the risk of severe disease, individuals with multiple comorbidities face substantially elevated risks, demonstrating the clinical significance of risk stacking in determining COVID-19 outcomes.

The national COVID-19 vaccine program (NCVP) was launched in 2021 by the Australian government to assist with the global impact of COVID-19 and provides free COVID-19 vaccines to all people in Australia [84]. COVID-19 vaccination is recommended for all people aged 18 years and older. All adults 75 years and older are recommended to have a COVID-19 vaccination every 6 months. All adults aged 65 to 74 years and adults aged 18 to 64 years who are severely immunocompromised are recommended to have a COVID-19 vaccination every 12 months, and can consider a dose every 6 months, based on their individual health needs. COVID-19 vaccination is strongly recommended for residents in aged care homes and aged care workers 18 years of age or older are encouraged to get a COVID-19 vaccination every 12 months [85]. All other adults aged 18 to 64 years can consider a dose of a COVID-19 vaccine every 12 months (see Appendix A) [3].

COVID-19 vaccination is also recommended for children aged 6 months to less than 18 years, with medical conditions that may increase their risk of severe disease or death from COVID-19. Unvaccinated pregnant women are recommended to receive the COVID-19 vaccine (Figure 1) [3].

COVID-19 vaccines have been very effective in preventing COVID-19 mortality in adults aged 65 years and older. In early 2022, absolute estimates of VE of a third booster dose received within 3 months following the previous vaccination were greater than 90% effective in preventing COVID-19 death. By late 2022, there was growing hybrid immunity due to widespread transmission of COVID-19, and estimates of the absolute VE were lower but still more than 80% [86]. A preprint by Liu et al. describes more recent analyses that demonstrated that the COVID-19 booster with variant-specific vaccine types including bivalent COVID-19 vaccines (ancestral and BA.1 or BA.4/5) and the monovalent COVID-19 XBB.1.5 vaccine were highly effective in preventing COVID-19 mortality [87]. Furthermore, a retrospective observational study between June and November 2022 observed that most nosocomial COVID-19 infections in a vulnerable hospital population were of mild severity and the mortality rate was low due to high vaccination coverage and readily available antiviral therapy [88]. However, an observational study of kidney transplant recipients in Australia found that overall seroconversion rates with three doses of vaccine were low (48%) in this population. Additionally, 30% of the cohort developed breakthrough COVID-19 [89].

Consistent with international data, Australian analyses show that COVID-19 VE decreases with increasing time since vaccine receipt [86,87]. Before the outbreak of COVID-19, there was evidence indicating that immunity to coronaviruses was generally short-lived and reinfection within 12 months was common. This implies that the decline in VE over time is not a result of the vaccine failing but is in line with a normal immune response [90]. Consequently, it is necessary to administer booster shots every 6 months to vulnerable populations, particularly older adults, who experience the most significant benefit of a reduction in mortality [91].

COVID-19 vaccination status varies by population groups in Australia. Coverage was high for aged care residents where 97% of residents had received three or more doses by 30 April 2022 [92]. However, by the end of 2024, only 71.9% of aged care residents had received vaccination in the past 12 months and there are many facilities with very low rates of vaccination coverage (less than 10%) [93,94]. By the end of April 2022, only 74% of National Disability Insurance Scheme (NDIS) participants and 53% of Aboriginal and Torres Strait Islander peoples over the age of 16 years had received three or more doses [92].

## 3. Expert Opinion

### 3.1. Gaps in Current Risk Group Definitions and Additional Risk Groups for Consideration

#### 3.1.1. Expanding Age Groups

There is still debate surrounding the ideal age to start vaccination for vaccine-preventable respiratory infections for adults, and in Australia the recommended ages for vaccination are different for PD, influenza, and RSV (Figure 1). Immunosenescence in older age leads to a decline in vaccine-induced immunity and an increased disease risk with age. Therefore, a trade-off is required between disease burden and vaccine efficacy by age [95]. The optimal age for vaccine recommendation is based on numerous factors including the incidence of the disease, the efficacy of vaccines, and cost-effectiveness modelling. The cost-effectiveness and the optimum age of vaccination are dependent on the timeliness of vaccine administration in the adult population, and it is important that models use realistic uptake profiles when assessing cost-effectiveness [95]. There has been an economic evaluation that has demonstrated favourable results in expanding the influenza vaccine to adults aged 50 to 64 years of age in Australia [96].

#### 3.1.2. Cochlear Implants or Cerebrospinal Fluid (CSF) Leaks

In Australia, the pneumococcal vaccine is recommended for people with proven or presumptive CSF leak, cochlear implants, and intracranial shunts due to their increased risk of developing pneumococcal meningitis [3]. Influenza and COVID-19 vaccinations are not currently recommended for these patients (Figure 1) and currently there are no studies linking cochlear implants or CSF leaks to severe influenza or COVID-19. Further research is required to investigate whether these conditions are generalisable risk factors for other VPRIs.

#### 3.1.3. Presence of Multiple Risk Factors (Risk Stacking)

Multimorbidity is common in Australia and tends to increase with age. In 2022, it was estimated that 38% of Australians had two or more long-term health conditions [97]. Studies have shown that there is an increased likelihood of developing pneumococcal disease for people with multiple concomitant risk factors (risk stacking) beyond the risk posed by individual risk factors. Furthermore, in an unvaccinated population aged 50 years and older, each individual risk factor increased the risk of mortality by 55% in the presence of PD highlighting the importance of vaccinating older individuals with multiple risk factors [98]. Studies have also shown a higher risk for COVID-19-attributable hospitalisation or death in those with more than two comorbidities [99]. An increasing number of high-risk comorbidities has also been associated with the potential for severe influenza outcomes [100].

#### 3.1.4. Frailty

The risk of infectious diseases in older adults has been found to increase with frailty, regardless of their age, sex, community activity participation, and exercise; an incidence rate ratio (IRR) of 1.97 (95% CI 1.44 to 2.71) was observed for respiratory-tract-transmitted diseases in frail older adults compared with healthy older adults [101]. Frailty has also been shown to increase the risk of developing COVID-19, influenza, and RSV [102,103,104]. Therefore, it is important to measure frailty and consider it as a potential risk group for vaccination.

#### 3.1.5. Aboriginal and Torres Strait Islander Peoples

The introduction of PCV13 (in 2011 to 2015) resulted in a 30% reduction in bacterial-coded pneumonia hospitalisation episodes among Aboriginal and Torres Strait Islander infants in the Northern Territory; however, the burden of acute lower respiratory infections (ALRIs) remains excessive in this population [105]. Targeted vaccination strategies may be less effective in this population due to dense and diverse colonisation by respiratory pathogens [105]. Several clinical trials have investigated novel vaccine schedules for PD in Aboriginal and Torres Strait Islander infants with limited success [6,106,107,108].

#### 3.1.6. High-Risk Paediatric Groups

While long term reductions in PD burden have been observed among most children following PCV use, the incidence of IPD was found to be largely unchanged, even after many years of high-population PCV coverage, for children with conditions associated with highest IPD risk, such as compromised immunity or CSF barrier. This may be due to expansion of non-vaccine types, and vaccines offering protection against a greater number of serotypes may be required for these children with high risk [16].

Despite the implementation of a standard vaccination program to prevent influenza in children following haematopoietic stem cell transplantation (HSCT), there is a high incidence of influenza infections which significantly contributes to morbidity [109]. Further research is required to understand how to prevent influenza infection in this high-risk population.

#### 3.1.7. High-Risk Adult Groups

Influenza continues to be a major cause of death and disease in high-risk populations indicating a need for additional research into more effective influenza vaccines. A randomised controlled trial was conducted to investigate the immunogenicity of a trivalent inactivated influenza vaccine (TIV) in adult patients with chronic disease (chronic lung disease, chronic heart disease, chronic renal disease, or diabetes mellitus) or those over 60 years of age. The study examined the TIV both alone or formulated with Advax delta insulin adjuvant. The combination had a positive effect on anti-influenza IgM and haemagglutination inhibition (HAI) levels; however, larger studies are required to determine how this might translate into improved clinical outcomes [110].

There is currently a significant lack of knowledge regarding the effectiveness of vaccines for respiratory infections in those aged over 85 years [111]. There is also uncertainty related to the effectiveness of these vaccines for people who are treated with an expanding array of immunosuppressive medications for autoimmune, rheumatological, and malignant conditions [112].

#### 3.1.8. People Living with Lung Diseases

People living with a lung disease are an at-risk group for VPRIs as indicated by the vaccination recommendations detailed in Appendix A. These VPRIs increase the likelihood of symptom exacerbation, deterioration in lung function, and death in this population, and therefore this group should be a priority for vaccination resourcing and targeting [113].

#### 3.1.9. Lifestyle Factors

While smoking is recognised as a significant independent risk factor for viral and bacterial respiratory infections, including severe influenza [3,114], it is not currently included as a specific criterion for NIP-funded vaccination in Australia (Figure 1). This represents a potential gap between identified risk factors and vaccination policy that warrants further consideration.

#### 3.1.10. Barriers to Vaccination Uptake

With the availability of effective vaccines against the major VPRIs, the principal challenge has shifted from vaccine development to vaccine implementation. Despite compelling evidence supporting vaccination efficacy, large gaps remain between potential and realised public health benefits. Understanding and addressing barriers to vaccination uptake represents a critical priority for improving population health outcomes.

##### Provider-Level Barriers

The COVID-19 pandemic has had an impact on vaccine uptake due to increased consumer confusion and distrust. Vaccinations have predominately been administered in general practice, which may be another barrier [113]. Vaccine uptake among healthcare professionals (HCPs) themselves is an important indicator of vaccine acceptance and a predictor of recommendation behaviour. Australian studies have demonstrated variable influenza vaccine coverage among HCPs. A systematic review found coverage rates ranging from 16.3% to 77% across different healthcare settings and professional groups, with physicians generally having higher uptake than nurses and allied health professionals [115]. Hospital-based HCPs typically achieve higher coverage (60–75%) than primary care settings (40–65%) [20]. Importantly, HCPs who are vaccinated themselves are significantly more likely to recommend vaccination to patients, creating a positive feedback loop. Despite this evidence, influenza vaccine coverage among Australian HCPs remains below optimal levels and well short of the 95% target recommended by professional bodies. Barriers to HCP vaccination include misconceptions about vaccine effectiveness and safety, concerns about adverse effects, belief in personal immunity, and practical access issues. Some studies have identified hesitancy even among physicians regarding certain vaccines, though this appears less pronounced in Australia compared to some international settings.

##### Healthcare Provider Attitudes and Vaccine Hesitancy

While healthcare providers are generally pro-vaccination, studies have identified a spectrum of attitudes ranging from strong advocacy to selective hesitancy or reluctance regarding specific vaccines or populations. Australian research provides important insights into this phenomenon. A 2019 Australian study examining antigen-specific vaccine hesitancy found that approximately 10–15% of healthcare providers expressed some hesitancy about recommending certain vaccines, with higher rates for newer vaccines (e.g., HPV, meningococcal B) compared to established childhood vaccines [116]. During the COVID-19 pandemic, provider attitudes evolved substantially. Initial hesitancy in 2021 (particularly regarding vaccination in pregnancy and young children) decreased significantly as safety and effectiveness data accumulated [117,118]. The impact of provider hesitancy on patient uptake can be substantial. Studies demonstrate that neutral or ambivalent recommendations (e.g., “it’s up to you” rather than obvious endorsement) result in significantly lower patient acceptance compared to strong, presumptive recommendations [115]. Provider hesitancy has been specifically linked to lower maternal influenza vaccination rates and reduced pneumococcal vaccine uptake in at-risk adults [119]. Contributing factors to provider hesitancy include incomplete knowledge about evolving recommendations, concerns about adverse effects in medically complex patients, previous negative patient experiences, time constraints limiting thorough vaccine discussions, and, rarely, philosophical objections to specific vaccines or vaccination policy. Addressing provider hesitancy requires continuing medical education emphasising current evidence, institutional support for vaccination activities, transparent communication about vaccine safety monitoring, peer-to-peer education programs, and clear, evidence-based guidelines that support confident recommendations. Professional medical organisations in Australia, including RACGP and RACP, have been active in providing such resources and advocating for strong provider vaccine recommendations [120,121].

##### Patient-Level Barriers

There are still significant obstacles to vaccinating older adults that are not present for younger age groups. These include difficulties accessing care, mobility limitations, multiple providers, lack of provider confidence in vaccines for older adults, lack of clinical trial data in older adults, and a predominantly paediatric immunisation culture [111].

##### Health-System Barriers

These barriers include out of pocket costs, confusion about which vaccines are free, and uncertainty about vaccine eligibility on the part of patients [113]. In Australia, it has been found that a major factor impacting coverage of the influenza vaccine during pregnancy was whether the vaccine was recommended by a doctor [116].

##### Strategies for Improving Implementation

Community pharmacies are increasingly playing a role in administering vaccination programs and serve as a convenient access point for vaccination services particularly for adults and those in remote locations [122]. The COVID-19 pandemic demonstrated that rapid implementation is achievable when multiple barriers are simultaneously addressed through expanded provider networks (pharmacists, nurses), clear public messaging, and removal of financial barriers [113]. Applying these lessons to routine VPRI vaccination could improve coverage. Pharmacists and GPs now offer vaccination clinics within residential aged care homes enabling improved vaccination coverage for this vulnerable population. The provision of maternal vaccines in hospital antenatal clinics has also been critical to improving rates of vaccine coverage in pregnant women [116]. Strategies to improve HCP vaccination rates includes workplace vaccination programs, peer education, addressing misconceptions, and institutional policies. These strategies may have dual benefits: directly protecting vulnerable patients from nosocomial transmission and indirectly improving patient vaccine confidence through stronger provide recommendations. Addressing these implementation challenges requires coordinated action across policy, health system, and community levels. Evidence-based strategies exist; the key challenge is systematic deployment and evaluation.

##### Strategies for Improving Implementation

Achieving equitable vaccination coverage requires proactive community engagement and culturally tailored promotional strategies, particularly for populations with access barriers or low baseline coverage. Australian governments have implemented various mass media campaigns promoting VPRI vaccination, including annual influenza campaigns, COVID-19 vaccination initiatives, and RSV awareness programs targeting pregnant women and older adults. These campaigns utilise television, radio, social media, and printed materials in multiple languages. Effectiveness data suggest that well-designed campaigns increase awareness and intent but must be coupled with access improvements to translate into actual uptake [123].

Recognising that generic campaigns often fail to reach Aboriginal and Torres Strait Islander communities effectively, targeted approaches have been developed including Aboriginal Community Controlled Health Organisations (ACCHOs) leading culturally safe vaccination programs, employment of Aboriginal Health Workers to provide trusted, culturally appropriate vaccine education community co-designed campaigns incorporating local languages, cultural protocols, and trusted community voices, and outreach clinics in remote communities timed to overcome seasonal access barriers for the integration of vaccination with other health services (e.g., chronic disease management, maternal health visits). Studies demonstrate that ACCHO-led programs achieve substantially higher coverage than mainstream services in the same populations [124].

While door-to-door vaccination campaigns are not routinely used in urban Australian settings, outreach approaches have been successfully deployed for remote and vulnerable populations. These include mobile vaccination clinics serving remote Indigenous communities, pharmacy-based “pop-up” vaccination services in community settings, workplace vaccination programs reaching populations who may not regularly access healthcare, aged care facility visiting vaccination programs, and school-based programs for eligible age groups (though focused on childhood/adolescent vaccines rather than VPRIs). The COVID-19 pandemic demonstrated the potential of intensive community outreach, with dedicated teams conducting home visits for homebound elderly and disability populations, achieving high coverage in these otherwise difficult-to-reach groups [125].

Lower vaccination coverage has been documented in some culturally and linguistically diverse (CALD) communities, attributed to language barriers, differing health beliefs, mistrust of healthcare systems, and access challenges [126]. Effective strategies include multilingual resources and interpreter services, community liaison workers from target communities, partnership with ethnic community organisations and faith leaders, targeted campaigns via ethnic media outlets, and cultural competency training for vaccinators.

While these strategies show promise, robust evaluation of promotional effectiveness specific to VPRI vaccination in Australian populations is limited. Particular gaps exist in understanding optimal approaches for newly arrived migrant communities, homeless populations, and other marginalised groups. Investment in both implementation and evaluation research is needed to build evidence for effective, equitable vaccination promotion strategies.

### 3.2. Limitations of This Review

This narrative review has several limitations. First, as a non-systematic review, we may not have captured all relevant evidence, particularly from grey literature sources. Second, the Australian focus means findings may not be fully generalisable to other healthcare contexts with different immunisation programs, disease epidemiology, or population demographics. Third, the rapidly evolving nature of vaccine recommendations, particularly for RSV and COVID-19, means that some information may become outdated. Fourth, limited Australian-specific data exist for some risk groups and vaccines, necessitating reliance on international evidence or modelled estimates. Finally, vaccine effectiveness estimates vary depending on study design, population characteristics, circulating strains, and outcome measures, making direct comparisons challenging.

## 4. Conclusions

This review highlights the substantial burden of PD, influenza, RSV, and COVID-19 for at-risk populations in Australia and demonstrates the clinical and economic value that vaccination provides. However, significant gaps between vaccine availability and uptake reveal critical opportunities for improving population health outcomes. Based on the evidence synthesised in this review, we present specific recommendations for Australian decision-makers across policy, health system, and clinical practice domains.

### 4.1. Policy-Level Recommendations

Harmonise age-based vaccination recommendations: Consider standardising the age threshold for adult vaccination across VPRIs where evidence supports alignment (e.g., 60 or 65 years), reducing complexity and potential confusion while maintaining disease-specific considerations where appropriate.Expand recognition of risk groups: Formally include validated risk factors currently absent from NIP criteria, particularly the following:
Smoking status for influenza vaccination eligibility;Frailty assessment for all VPRI vaccinations in older adults;Explicit recognition of risk stacking (multiple comorbidities) as a distinct high-priority category.Address financial barriers: Review cost-sharing arrangements for at-risk populations currently requiring co-payment, particularly for pneumococcal vaccination in adults under 70 years with medical risk factors, where uptake is notably low.Strengthen Aboriginal and Torres Strait Islander programs: Increase investment in ACCHO-led, culturally appropriate vaccination programs with demonstrated-superior outcomes in these communities. Consider lower age thresholds for routine adult vaccination reflecting earlier disease onset.Enhance surveillance: Improve collection and reporting of the following:Vaccine coverage data stratified by risk group, particularly medical comorbidity status;Disease burden in at-risk populations;Vaccine effectiveness in immunocompromised and elderly populations.

### 4.2. Health System Recommendations

Expand vaccination provider networks: Formalise and expand roles for pharmacists, nurses, and Aboriginal Health Workers in VPRI vaccination, building on COVID-19 pandemic successes.Implement systematic identification: Develop electronic medical record prompts and decision support tools to systematically identify eligible patients, particularly those with multiple risk factors.Coordinate respiratory vaccine delivery: Establish coordinated “respiratory vaccine programs” allowing eligible adults to receive multiple indicated vaccines in planned visits, improving convenience and coverage.Target aged care facilities: Mandate regular vaccination audits and improvement programs for residential aged care, where recent coverage decline poses significant risk.Improve healthcare worker vaccination: Implement workplace programs targeting HCP vaccination, recognising the dual benefits of workforce protection and improved patient recommendations.

### 4.3. Clinical Practice Recommendations

Adopt presumptive recommendations: Train healthcare providers to use strong, presumptive vaccine recommendations (“You are due for your flu shot today”) rather than passive offers, which have demonstrated higher acceptance rates.Assess risk stacking: Systematically identify patients with multiple VPRI risk factors who warrant prioritisation for all eligible vaccinations.Address vaccine hesitancy: Provide training and resources for providers to confidently address common concerns and hesitancy regarding VPRI vaccines.Leverage opportunistic moments: Utilise all clinical encounters (chronic disease management, hospital discharge, specialist visits) as vaccination opportunities.

Research Priorities:Optimal age for vaccination: Conduct Australian-specific cost-effectiveness analyses incorporating realistic uptake scenarios to determine optimal age thresholds, particularly for pneumococcal and RSV vaccines.Vaccine effectiveness in high-risk groups: Address critical evidence gaps regarding VE in immunocompromised populations, frail elderly (>85 years), and patients on immunosuppressive therapies.Implementation science: Evaluate effectiveness of different promotional strategies, provider interventions, and health system approaches in diverse Australian populations.High-risk paediatric populations: Develop and evaluate enhanced vaccine schedules or formulations for children with compromised immunity, particularly post-HSCT, where standard approaches show limited effectiveness.Risk prediction tools: Develop and validate practical risk stratification tools to identify individuals who would benefit most from vaccination beyond current categorical eligibility criteria.

### 4.4. Concluding Statement

The foundation for preventing VPRI burden in at-risk Australians exists: safe, effective vaccines are available and funded for most high-risk groups. The challenge—and opportunity—lies in closing implementation gaps through coordinated policy, health system innovation, and clinical practice improvements. By adopting these evidence-based recommendations, Australia can realise substantial gains in health outcomes, health equity, and healthcare system sustainability. The time to act is now, before the next respiratory disease season.

## Figures and Tables

**Figure 1 vaccines-13-01212-f001:**
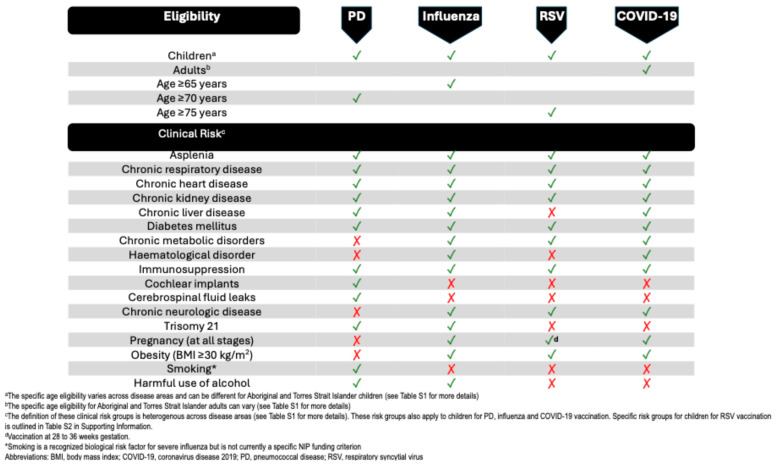
Australian adult and child vaccination recommendations for vaccine-preventable respiratory infections. ✓ indicates NIP-funded vaccination eligibility, × indicates not NIP eligible; biological risk factors may exist beyond funded criteria.

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
