# Peer review of "Risk Groups for Vaccine-Preventable Respiratory Infections in Children and Adults: An Overview of the Australian Environment"

_vaccines, 2025, doi:10.3390/vaccines13121212_

Round 1
Reviewer 1 Report
Comments and Suggestions for Authors
The paper effectively synthesizes the complex and often fragmented immunisation guidelines for pneumococcal disease (PD), influenza, RSV, and COVID-19 within the Australian context. The authors have done an excellent job of not only outlining the current recommendations but also delving into the epidemiological burden, vaccine effectiveness, and, critically, the persistent gaps in both policy and research.
However, some revisions are needed to improve methodological transparency, strengthen the neutrality of the narrative, and clarify several interpretations.
- The concept of "risk stacking" (i.e., multimorbidity) is introduced in the abstract and explored well in the Discussion (Section 4.1.3). This is arguably one of the most important clinical insights of the review. This concept could be strengthened by integrating it earlier and more explicitly into the disease-specific sections (Section 3).
- Section 4.1.10 does a good job listing potential barriers (cost, confusion, provider recommendation, etc.). However, this point could be elevated. Given that effective vaccines exist for most of these conditions, the key challenge is now one of implementation.
- Limitation section needs, in order to strengthen the methodological transparency, I suggest adding a brief statement to the limitations (either in the discussion or at the end of the methods paragraph) that explicitly acknowledges this.
- In Figure 1, "Smoking" is marked with a "✓" for PD but an "X" for Influenza, RSV, and COVID-19. However, the text in Section 4.1.9 rightly points out that smoking is a significant risk factor for severe influenza, even if it's not currently a specific criterion for NIP-funded vaccination. This creates a slight contradiction.
- In Section 3.1.2, the paper notes that VE estimates for influenza ranged from 13% to 52%. While the text mentions factors influencing VE, this wide range might lead some readers to question the vaccine's value.
-
The manuscript states (Section 3.1.3) that there is a "lack of data on RSV disease burden in older adults in Australia," but then provides a very high modelled hospitalisation rate (256 per 100,000 for those >75). This is a slight contradiction. I suggest rephrasing to state that while systematic surveillance data has been lacking (as it's not notifiable, etc.), recent modelling studies and increased testing are revealing a substantial and previously under-appreciated burden, which justifies the new vaccine recommendations.
Reviewer 2 Report
Comments and Suggestions for Authors
Comments and Suggestions for Authors
In this review, the authors provide a comprehensive overview of vaccine-preventable respiratory infections (VPRIs) and summarise the current adult and paediatric immunisation guidelines for VPRIs in Australia. They evaluate the burden of VPRIs, vaccine efficacy, vaccine recommendations for high-risk groups, and vaccination coverage. The manuscript highlights gaps in existing definitions of high risk groups and discusses additional groups that could be considered for inclusion in future vaccine recommendations. The study presents important findings in a clear, systematic manner and offers conclusions that may assist Australian public health authorities in shaping appropriate vaccination policy strategies. However, I have several essential comments that could help improve the quality of the manuscript. These are outlined below:
Comments:
Sections 1-3.
- There is considerable repetition throughout the manuscript (e.g., lines 162–171). I therefore recommend substantially condensing and shortening the more general and widely known sections of the text, particularly those describing the diseases and their causative agents.
-Lines 79-81: For readers who may not be familiar with Aboriginal and Torres Strait Islander ethnicity groups, please provide an explanation of the factors contributing to the higher prevalence and burden of VPRIs in these communities.
- Lines 135-142: What are the recent pneumococcal vaccination coverages in Aboriginal and Torres Strait Islander populations?
-Lines 211: When reporting vaccine effectiveness, always specify the outcome to which it refers (e.g., infection, hospitalization, or death) because the effectiveness can vary significantly for each outcome.
-Lines 262-263: I suggest to omit references that cite older data (more than 10 years ago).
Line 354: I recommend reporting rates instead of absolute numbers, for example, using the mortality rate rather than the number of people who died from COVID-19.
Line 458: I suggest omitting the phrase "respiratory vaccination".
Section 4. Expert Opinion
-Line 548-550: In Australia, it was found that the main factor influencing influenza vaccination coverage during pregnancy was whether the vaccine was recommended by a doctor. However, it would be valuable for the authors to also discuss influenza vaccine uptake among healthcare professionals themselves, including physicians, as this could serve as an important indicator of their acceptance of the vaccine and, consequently, their likelihood of recommending it.
-The authors should also address potential reluctance among treating physicians to recommend vaccination against VPRIs. Are there indications that some healthcare providers hold anti-vaccination attitudes, and could this have influenced the vaccine uptake? If relevant studies on this issue have been conducted in Australia, their findings should be included and discussed, along with the appropriate references.
-What about vaccination promotion, media campaigns and door-to-door education, especially for some remote ethnic groups where coverage is very low?
Section 5. Conclusion
The conclusion is too general. Based on your findings, do you have any specific recommendations for decision-makers in Australia that could be drawn from this review?
